# Discrimination between Obstructive Coronary Artery Disease and Cardiac Syndrome X in Women with Typical Angina and Positive Exercise Test; Utility of Cardiovascular Risk Calculators

**DOI:** 10.3390/medicina55010012

**Published:** 2019-01-14

**Authors:** Mohammad Saadat, Farzad Masoudkabir, Mohsen Afarideh, Saeed Ghodsi, Ali Vasheghani-Farahani

**Affiliations:** 1Cardiac Primary Prevention Research Center, Tehran Heart Center, Tehran University of Medical Sciences, Tehran 1411713138, Iran; saadatagah.m@gmail.com (M.S.); farzad.masoudkabir@gmail.com (F.M.); dsaeedgh@gmail.com (S.G.); 2Department of Cardiology, Tehran Heart Center, Tehran University of Medical Sciences, Tehran 1411713138, Iran; mhafarideh@gmail.com; 3Department of Electrophysiology, Tehran Heart Center, Tehran University of Medical Sciences, Tehran 1411713138, Iran

**Keywords:** exercise tolerance test, cardiac syndrome x, coronary angiography, 10 years risk of ASCVD, Framingham risk score, euro score

## Abstract

*Introduction:* Nearly 40% of women with typical angina and a positive exercise tolerance test (ETT) have normal or near normal coronary angiography (CAG) labeled as cardiac syndrome X (CSX). *Objective:* We performed this study to evaluate the power of common cardiovascular risk calculators to distinguish patients with CSX from those with coronary artery disease (CAD). *Methods:* 559 women participated in the study. Three risk scores, including (1) newly pooled cohort equation of American College of Cardiology/American Heart Association (ACC/AHA) to predict 10 years risk of first atherosclerotic cardiovascular hard event (ASCVD), (2) Framingham risk score (FRS) for the prediction of 10 years coronary heart disease, and (3) the SCORE tool to estimate 10-year risk of cardiovascular mortality (SCORE), were applied. *Results:* CAD was diagnosed in 51.5% of the patients. 11.6% of the population had ASCVD < 2.5%, and only 13.8% of these patients had CAD on their CAG. By choosing FRS, 14.4% of patients had FRS < 7.5%, and only 11.3% of these patients had recorded CAD on CAG, while the rest of the patients were diagnosed as CSX. Using the SCORE model, 13.8% of patients had the least value (<0.5%) in whom the prevalence of CAD was 19.9%. The area under receiver operating characteristic curve (AUROC) to discriminate CSX from CAD was calculated for each scoring system, being 0.750 for ASCVD, 0.745 for FRS, and 0.728 for SCORE (*p* value for all AUROCs < 0.001). The Hosmer–Lemeshow chi squares (df, *p* value) for calibration were 8.787 (8, 0.361), 11.125 (8, 0.195), and 10.618 (8, 0.224) for ASCVD, FRS, and SCORE, respectively. *Conclusions:* Patients who have ASCVD < 2.5% or FRS < 7.5% may be appropriate cases for noninvasive imaging (Such as coronary CT angiography). CAG is indicated for patients with ASCVD ≥ 7.5% and FRS ≥ 15%, whereas the patients with intermediate risk need comprehensive patient–physician shared decision-making.

## 1. Introduction

In order to evaluate patients with chest pain, more than 2 million and more than 1.7 million orders of coronary angiography (CAG) are recorded annually in Europe [1] and USA [2], respectively. Several studies have shown that 10% to 30% of patients undergoing CAG have normal or near normal coronary arteries [1]. Cardiac syndrome X (CSX) is defined as typical chest pain accompanied by both a positive exercise tolerance test (ETT) and non-obstructive CAG patterns [3,4,5,6,7]. To date, numerous studies have been performed to explain the underlying mechanisms of CSX. Several possible mechanisms, including but not limited to microvascular dysfunction, platelet hyperactivity, and lactate formation, have been suggested [3,4,5,8,9,10,11,12,13].

CSX is acknowledged as a female-predominant disorder [14]. Interestingly, in a large cohort study of patients suspected to have myocardial ischemia, 41% of women versus only 8% of men had normal or near-normal coronary arteries in CAG [15]. Prediction of CAG results has been an interesting and challenging topic in cardiology practice for a long time. Previous investigations have shown lower prevalence rates of cardiovascular risk factors among patients with CSX compared with the CAD group [16,17,18]. Thus, several studies have proposed risk prediction models for obstructive CAD in CAG [19,20,21,22,23].

Multiple, well-regarded scoring systems were developed to predict the absolute risk of future cardiovascular events by using major risk factors. This cross-sectional study was conducted to evaluate the performance of these risk calculators to distinguish patients with CSX from those with CAD in women with typical angina and a positive ETT. For this purpose, the following risk scores were applied for each patient. The first model was a newly pooled cohort equation of ACC/AHA to predict the ten years risk of the first atherosclerotic cardiovascular hard event (ASCVD) [24]. The second model was the Framingham risk score (FRS) for the prediction of ten years risk of coronary heart disease [25]. The third one was the SCORE project introduced by European Society of Cardiology to estimate 10-year risk of cardiovascular mortality (SCORE) [26].

Although these risk scores help to assess different clinical outcomes, we aimed to examine their efficiency in the prediction of CAG findings. For example, CHADS_2_ and CHA_2_DS_2_-VASc are two validated scoring systems initially introduced to determine the risk of stroke in patients with non-valvular atrial fibrillation (AF) [27,28]. Subsequent studies addressed their advantage in the prediction of the risk for the development of new-onset AF [29,30]. Recent studies support the role and benefit of these models in predicting the prognosis of myocardial infarction (MI) [31,32,33] and patients after coronary intervention [34].

These models are valuable tools in clinical practice through paper scoring sheets, websites, and mobile applications. Hence, the risk calculators have become integral parts of the primary prevention and decision-making programs [24].

An accurate prediction of CAD probability would greatly reduce the imposed burdens and costs of unnecessary therapeutic interventions [2]. This would also have a direct impact on the physician–patient shared decision-making. Moreover, a non-invasive modality would be the appropriate choice for a large proportion of patients who have low pretest probability for CAD (suspected for CSX).

## 2. Methods

We carried out this study using the data from the ongoing database of Tehran Heart Center (Tehran University of Medical Sciences, Tehran, Iran). Comprehensive information profiles of the patients were achieved, including demographic data, symptoms, past medical history, risk factors and comorbidities, glucose and lipid profile, electrocardiogram findings, echocardiographic parameters, results of their myocardial perfusion scan, exercise tolerance test, and coronary catheterizations. In our center registry, all variables were defined and measured according to the American College of Cardiology key data elements and definitions for measuring the clinical management and outcomes of patients with acute coronary syndromes [35]. From 2004 until 2014, all patients with typical angina and a positive exercise tolerance test who underwent diagnostic coronary angiography at our center were enrolled in the study. Initially, those who had undergone CAG because of emergency conditions such as acute coronary syndrome were excluded. The presence of a previous history of myocardial infarction, percutaneous coronary intervention, coronary artery bypass grafting, and ejection fraction (EF) <50% were considered as other exclusion criteria.

Initially, we determined the frequency of outcomes (CAD and CSX) in both genders. A preliminary analysis of data revealed that 85.7% of male patients with typical angina and positive ETT had CAD on their coronary angiography report. Hence, performing CAG in these patients is a substantially appropriate choice. Conversely, a significantly lower proportion of female patients (55.65%) with typical angina and positive ETT had positive CAG findings. In a similar manner, 77% of women who had highly positive test results had angiographic CAD; therefore, in the final analysis, this group was also excluded. For this purpose, study subjects were restricted to female patients with positive ETTs (not highly positive). Ultimately, 559 women met the eligibility criteria for final analysis.

Patients with atherosclerotic plaques leading to 50% or more luminal stenosis in one or more of the major epicardial vessels were considered as obstructive CAD cases. Patients with mild CAD (<50%) or completely normal coronary vasculature were identified as being the cases of CSX.

ETT was considered positive if the patient had one or more of the following criteria: (I) both ischemic discomfort and ST shift ≥1 mm (0.1 mV, horizontal or down sloping) or (II) new ST shift ≥2 mm (0.2 mV, horizontal or down sloping) despite the absence of ischemic discomfort. Highly positive ETT is defined as one or more of the following: (I) ST depression in 5 or more of the precordial leads; (II) maximal ST depression >2 mm; (III) a positive test with a heart rate <120; (IV) hypotension during exercise; (V) exercise capacity <5 min [36].

All patients had declared their informed consent about the total process of clinical management in addition to the use of their available data in research programs. Furthermore, the present study received the approval of Tehran University of Medical Sciences ethics committee addressed by IR.TUMS.MEDICINE.1395.4760 code.

## 3. Statistical Analysis

ASCVD, FRS, and SCORE risk prediction models were calculated for each patient. With regard to the mathematical nature of the models, the absence of data on even one of the risk factors impeded the calculation of individual risk scores. As a result, 162 patients had at least one missing value or incomplete data of variables required for calculations, and consequently, computed scores of 559 patients were included in the final analysis.

Area under the receiver operating characteristic curves (AUROC) were employed to assess the discriminatory power of various risk-scoring scales to differentiate patients with CSX from those with obstructive CAD in CAG. Goodness-of-fit testing (Hosmer–Lemeshow chi square) was used to assess how well the model was calibrated. To obtain a simple algorithm to predict the likelihood of CSX, patients were classified according to their earned scores in each model through different ways, and finally, the best model of risk stratification was selected. In addition, multivariate logistic regression analysis was performed to demonstrate the association between obstructive CAD and these risk scores. Thus, odds ratios with 95% confidence intervals were recruited.

All the statistical analyses were accomplished with the use of the commercially available package SPSS version 21.0 (SPSS Inc., Chicago, IL, USA).

## 4. Results

We enrolled 559 female participants for the final analysis step. Of these participants, 288 patients were found to have obstructive CAD (51.5%) and 271 patients (48.5%) had CSX. The mean (±SEM) age was 56.16 ± 0.35 years for the whole population and 59.01 ± 0.45 years against 53.14 ± 0.48 years for the CAD and CSX subgroups, respectively (*p* value < 0.001). As shown in Table 1, the prevalence of diabetes, dyslipidemia, hypertension, and smoking were significantly higher in the CAD group than in the CSX patients (*p* value for all < 0.001). Table 1 depicts baseline demographic, clinical, and laboratory characteristics of the study patients. The AUROC (95% CI) was calculated for each scoring system, being 0.750 (0.710–0.790) for ASCVD, 0.745 (0.704–0.785) for FRS, and 0.728 (0.687–0.770) for SCORE (*p* value for all AUROCs< 0.001) (Figure 1). Hosmer–Lemeshow chi square values (df, *p* value) for calibration were 8.787 (8, 0.361), 11.125 (8, 0.195), and 10.618 (8, 0.224) for ASCVD, FRS and SCORE, respectively.

Table 2 shows the probability of CSX versus CAD pertaining to different risk categories of the three aforementioned scoring models (low, intermediate, and high pretest probability). Of the population, 11.6% (95% CI: 8.9%–14.3%) belonged to the low-risk category of ASCVD (predicted pretest risk of less than 2.5%), with only 13.8% (95% CI: 5.4%–22.2%) of these patients having CAD on their CAG report. In fact, 86.2% of women with low ASCVD risk (95% CI: 77.8%–94.6%) were diagnosed as CSX. 58.7% (95% CI: 54.6%–62.8%) of patients had an ASCVD risk >7.5% (high pretest probability of CAD), with a considerable number of them (67.1% 95% CI: 62.0%–72.2%) recording CAD on their CAG.

By choosing a cut-off point of less than 7.5% to indicate low-probability FRS risk, 14.4% (95% CI: 11.4% to 17.2%) of women were tagged as low-risk subjects. However, only 11.3% of these patients (95% CI: 4.4% to 18.2%) had recorded CAD on CAG, with the rest of these patients [88.7% (95% CI: 81.8% to 95.6%)] being cases of CSX. We found that 61.8% (95% CI: 57.8% to 65.8%) of the patients had FRS more than 15% among whom the prevalence of CAD was 66.4% (95% CI: 61.4% to 71.4%).

Using the SCORE calculator, 13.8% (95% CI: 10.9% to 16.7%) of patients had SCORE less than 0.5% (labeled as low risk group) and corresponding prevalence of CAD was 19.9% (95% CI: 10.7% to 28.4%). Finally, 63.3% (95% CI: 59.6% to 67.6%) of patients obtained SCORE risk of more than 1.5% (indicating high-risk group), with CAD prevalence of 65.3% (95% CI: 60.3% to 70.3%).

We found 70 patients (12.5%) under 45 years old. Among these low risk patients, 31 (48.4%) and 34 (53%) patients had ASCVD less than 2.5% and FRS less than 7.5%, respectively. Surprisingly 96.8% of these patients with ASCVD < 2.5 percentage and 97.1% of these patients with FRS < 7.5 percentage had CSX instead of CAD.

Results of Table 3 indicate to multivariate association of scoring models with obstructive CAD with respect to increased predicted risk. For instance, individuals who had moderate and high-risk profile according to ASCVD attained Odds ratios about 14.5 and 3.7 (compared with low risk category). We also found significant interdependences pertaining to the other models. A subgroup analysis for women according to diabetes showed that high SCORE points (risk ≥ 1.5% versus < 0.5%) are associated with greater likelihood of CAD. Corresponding odds ratios were 11.42 (2.54–16.71) and 5.53 (1.43–8.92) for diabetics and non-diabetics respectively. Comparison of intermediate and low risk patients ((0.5–1.5%) against <0.5%) did not lead to statistical significant results.

## 5. Discussion

The ACC/AHA integrated guideline on the management of stable ischemic heart disease (SIHD) recommends CAG in patients whose clinical characteristics or results of noninvasive testing implies high pretest probability of CAD (>90%). Coronary angiography has been also suggested for intermediate-risk patients who have an unsatisfactory quality of life due to angina with left ventricular ejection fraction of 50% or more [37]. Women with typical chest pain who were under 60 years of age have been considered as intermediate-risk, while the others (>60 years) were tagged as high-risk individuals. As a result, these patients will be candidates for CAG if they had a positive noninvasive test, such as ETT.

As shown in our study and previous publications [38], nearly 50% of women with those attributes will have normal epicardial coronary arteries or mild CAD on their coronary angiograms. Based on these shortcomings, we believe that algorithms that are more detailed will be required to establish the method of choice from noninvasive imaging and CAG for a state-of-the-art practice in each clinical situation. Appropriate Use Criteria for Diagnostic Catheterization [39] recommends applying FRS to determine pretest probability of CAD, but the exact role of the RFS and other risk estimation models in this setting remains to be elucidated. However, this guideline emphasizes that commonly used risk scores should be validated for certain populations (e.g., women, younger men, minority populations). Hence, further evaluation in these populations would be required. Although some studies reported low accuracy and overestimation in these models [40,41,42], this issue was not important in our study. Indeed, what was important is true classification and stratification of patients rather than their absolute predicted risk.

In order to detect CAD, CT angiography has optimal sensitivity and a high negative predictive value [43,44,45]. It has been demonstrated that the appropriate use of CTA reduces the rate of normal angiographic findings on CAG [46]. Based on these observations, the ACC/AHA integrated guideline on SIHD suggests CTA for patients with low to intermediate pretest probability of CAD [37], whereas performing CTA in patients with a high likelihood of CAD is often inappropriate [47]. The methods to define the pretest probability of CAD remain a challenge. As shown in our study and many other studies, the simple method of Appropriate Use Criteria for Diagnostic Catheterization [39], which uses just age and pain characteristics, had low utility in the enrollment of patients for CAG.

The relatively simple and practical stratification of patients in our study by means of common cardiovascular risk calculators led to a better prediction of CAD probability. Among the calculated scales, FRS and ASCVD performed largely superior to SCORE. One of the most important factors to assess these models was a lower prevalence of CAD in patients labeled as low-risk. Among patients who were tagged as low-risk based on FRS, 11.3% had CAD on their CAG. This proportion was 13.8% for ASCVD and 19.5% for SCORE. Therefore, FRS was superior to ASCVD because FRS put 14.3% of patients in the low-risk category with the likelihood of 11.3% for CAD, whereas ASCVD stratified 11.6% of patients in the low-risk group with 13.8% of cases being positive for CAD. Taken together, a higher percentage of patients would stay CAG-free by applying FRS as a valuable screening tool.

In contrast to our results, “Kones “stated that 95% of females who had their first myocardial infarction were assigned to the low-risk category based on Framingham Risk Scores, with the remaining 5% placed in the intermediate group [48]. Similar to our results, D’Agostino and colleagues demonstrated that the sex-specific risk prediction was relatively effective, with the C-statistics power being 0.79 [0.77 to 0.81] in women [49]. The majority of the previous reports have confirmed the usefulness of FRS, SCORE, and ASCVD scores regarding the calculation of CAD risk among women [24,50,51,52,53]. Our new findings were the partial confounding role of the diabetes, which reduces the effect of high SCORE on CAD. Albeit we applied the median and quartile values instead of traditional cut-off points, the validation seems suboptimal.

This study bears several important limitations. First, the single-center nature of this study precludes generalization of the results to patients with CSX from other institutions, ethnic groups and countries. Additionally, and due to a variety of reasons (e.g., the lack of highly sensitive C-Reactive protein (hs-CRP) measurement in our database), we were unable to conduct further comparisons with other common risk-scoring systems, including the Reynolds Score [54] that was recently demonstrated to be a superior risk-scoring system compared to both ASCVD and FRS [41,42].

## 6. Conclusions

Our study implemented the newly pooled cohort equation of ACC/AHA (ASCVD) and FRS to determine the pretest probability of CAD in women with typical chest pain and positive ETT. Patients who have ASCVD < 2.5% and those who have FRS < 7.5% may be appropriate cases for CTA instead of CAG. However, we must consider the crucial principles of a clinical approach to patients with chest discomfort. These scoring models are secondary tools which may help decision-making. Thus, in women who have clinical features such as warning ECG changes and typical recurrent refractory unstable angina are suitable for CAG despite having low scores. CAG remains the logical choice for patients with ASCVD ≥ 7.5% and FRS ≥ 15%. However, we suggest borderline patients between these risk categories to undergo a comprehensive plan of shared decision-making comprised of weighing up clear benefits offered by the use of CAG against potential hazards and adverse events and the probability of positive results in CAG.

In conclusion, our proposed algorithm might present a useful yet simple practical tool for clinicians to distinguish between women with CSX and those with obstructive CAD.

## Figures and Tables

**Figure 1 medicina-55-00012-f001:**
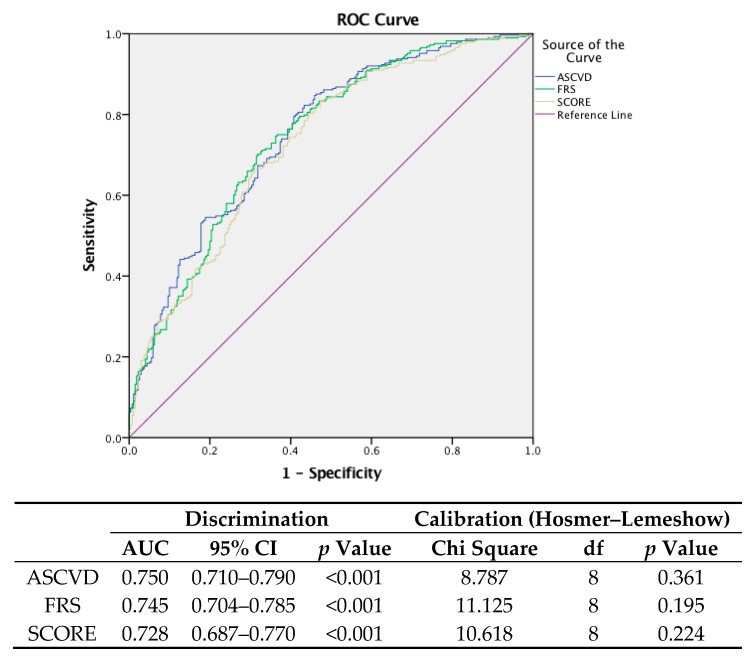
Receiver operating characteristic (ROC) curve to discriminate CSX from CAD.

**Table 1 medicina-55-00012-t001:** Baseline demographic, clinical, and laboratory characteristics of the study patients.

	CAD (288) (51.5%)		CSX (271) (48.5%)		Total (559) (100%)		*p* Value
Age (Years)	59.01	(0.45)	53.14	(0.48)	56.17	(0.35)	<0.001
DM, *n* (%)	110	(38.2%)	57	(21.0%)	167	(29.9%)	<0.001
HTN, *n* (%)	202	(70.1%)	129	(47.8%)	331	(59.3%)	<0.001
CS, *n* (%)	208	(72.7%)	162	(59.8%)	370	(66.2%)	<0.001
FH, *n* (%)	80	(28.2%)	64	(23.7%)	144	(26.0%)	NS
BMI	28.89	(0.26)	29.05	(0.28)	28.97	(0.19)	NS
EF (%)	58.69	(0.37)	59.19	(0.33)	58.93	(0.25)	NS
SBP (mmHg)	153.65	(1.35)	144.49	(1.25)	149.20	(0.94)	<0.001
DBP (mmHg)	80.24	(0.58)	78.00	(0.58)	79.15	(0.41)	0.006
T.Cho (mg/dL)	208.74	(2.82)	195.04	(2.17)	202.10	(1.93)	<0.001
TG (mg/dL)	181.03	(5.26)	157.30	(4.93)	169.52	(3.64)	<0.001
LDL (mg/dL)	127.85	(2.57)	118.23	(2.17)	123.19	(1.70)	0.005
HDL (mg/dL)	47.92	(0.59)	48.91	(0.71)	48.40	(0.46)	NS
Cr (g/dL)	0.92	(0.01)	0.86	(0.01)	0.89	(0.01)	NS
FBS (mg/dL)	127.24	(3.40)	112.25	(2.47)	119.98	(2.14)	<0.001

Quantitative variables represented as value (SEM); BMI: Body Mass Index, CAD: Coronary Artery Disease, CS: Cigarette Smoking, Cr: Creatinine, CSX: Cardiac Syndrome X, DBP: Diastolic Blood Pressure, DM: Diabetes Mellitus, EF: Ejection Fraction, FBS: Fasting Blood Sugar, FH: Family History, HDL: High Density Lipoprotein, HTN: Hypertension, LDL: Low Density Lipoprotein, SBP: Systolic Blood Pressure, T.Chol: Total Cholesterol, TG: Triglyceride.

**Table 2 medicina-55-00012-t002:** Probability of CSX versus CAD in three different risk categories.

		CAD		CSX		Total	
ASCVD Risk (New Pooled Cohort Equation)	Less than 2.5%	13.8%		86.2%		65	(11.6%)
		(5.42%–22.18%)		(77.82%–94.58%)		(8.95%–14.25%)	
	2.5% to 7.5%	35.5%		64.5%		166	(29.7%)
		(28.22%–42.78%)		(57.22%–71.78%)		(25.91%–33.49%)	
	7.5% or More	67.1%		32.9%		328	(58.7%)
		(62.02%–72.18%)		(27.82%–37.98%)		(54.62%–62.78%)	
Framingham Risk Score	Less than 7.5%	11.3%		88.7%		81	(14.4%)
		(4.26%–18.24%)		(81.76%–95.64%)		(11.40%–17.20%)	
	7.5% to 15%	37.6%		62.4%		133	(23.8%)
		(29.27%–45.73%)		(54.17%–70.63%)		(20.27%–27.33%)	
	15% or More	66.4%		33.6%		345	(61.8%)
		(61.42%–71.38%)		(28.62%–38.58%)		(57.77%–65.83%)	
Euro Score	Less than 0.5%	19.5%		80.5%		77	(13.8%)
		(10.65%–28.35%)		(71.65%–89.35%)		(10.94%–16.66%)	
	0.5% to 1.5%	32.8%		67.2%		128	(22.9%)
		(24.67%–40.93%)		(59.07%–75.33%)		(19.42%–26.38%)	
	More than 1.5%	65.3%		34.7%		354	(63.3%)
		(60.34%–70.26%)		(29.74%–39.66%)		(59.61%–67.59%)	
Total		288	(51.5%)	271	(48.5%)	559 (100%)	
		(47.36%–55.64%)		(44.36%–52.64%)			

**Table 3 medicina-55-00012-t003:** Association of CAD with various cardiovascular risk scores.

Association with CAD		Odds Ratio (95% Confidence Interval)		*p*-Value
ASCVD	<2.5%	Reference category		ref
	2.5–7.5%	3.694	(2.495–5.470)	<0.001
	≥7.5%	14.514	(6.687–31.503)	<0.001
FRS	<7.5%	Reference category		ref
	7.5–15%	3.277	(2.162–4.967)	<0.001
	≥15%	17.767	(8.277–38.137)	<0.001
SCORE	<0.5%	Reference category		ref
	0.5–1.5%	3.846	(2.504–5.905)	<0.001
	≥1.5%	8.451	(4.551 -15.695)	<0.001

ASCVD: atherosclerotic cardiovascular hard event, FRS: Framingham risk score, SCORE: 10-year risk of cardiovascular mortality.

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
