# Peer review of "Discrimination between Obstructive Coronary Artery Disease and Cardiac Syndrome X in Women with Typical Angina and Positive Exercise Test; Utility of Cardiovascular Risk Calculators"

_medicina, 2019, doi:10.3390/medicina55010012_

Round 1

Reviewer 1 Report

The manuscript is overall well written with results having clinical value and impact. A few minor comments are listed below for improvement:

Line 48-49. few studies- a few studies. 

Line 56, (25) and full stop "." is separated. 

Line 63, why the full stop after ref 34 is in red colour?

Line 119, a total of-A total of.

Line 138, there is more space between CAD and pertaining. Please double check and fix the error. 

Line 147, by choosing a cut point-by choosing a cut-off point. 

Line 171: .(37) change to (37).

Line 191, (47).,-(47). the challenge-The challenge.

Discussion: Authors need to highlight how their study compares to others in the literature and emphasise the significant contribution of findings from this study. 

Author Response

Applied corrections:

1.a few studies.

2. The first model was a newly pooled cohort equation of ACC/AHA to predict 10 years risk of first atherosclerotic  cardiovascular hard event (ASCVD) (24). Second model was the Framingham risk score (FRS) for the prediction of 10 years coronary heart disease (25)  .Third one was the SCORE project introduced by European Society of Cardiology to estimate 10-year risk of cardiovascular mortality (SCORE) (26).

3. the full stop after ref 34 is in red colour was changed to black.

4. A total of :  the sentence was changed asWe enrolled 559 female participants for the final analysis step. 

5. Line 138, there is more space between CAD and pertaining. Please double check and fix the error :  The gap was reduced properly.

6. by choosing a cut-off point.

7..(37) changed to (37).

8.Line 191, (47).,-(47). the challenge- changed to The challenge

9.comparison of the results with prior studies:

In contrast to our results, “Kones “stated that 95% of females, who had their first myocardial infarction, were assigned to the low risk category based on Framingham Risk Scores, with the remaining 5% placed in the intermediate group (48). Similar to our results, “D, Agostino” and colleagues demonstrated that the sex-specific risk prediction was relatively good, the C-statistics power was 0.79, [0.77 to 0.81] in women (49). Majority of the previous reports have confirmed the usefulness of FRS, SCORE, and ASCVD scores regarding calculation of CAD risk among women (24, 50-53). Our new findings were the partial confounding role of the diabetes, which declines the effect of high SCORE on CAD. Albeit we applied the median and quartile values instead of traditional cut-off points , the validation seems suboptimal.  

This study bears several important limitations. First, the single-center nature of this study precludes generalization of the results to patients with CSX from other institutions, ethnic groups and countries. Additionally and due to a variety of reasons (e.g., the lack of hsCRP measurement in our database), we were unable to conduct further comparison with other common risk scoring systems, including the Reynolds Score (54) that recently introduced itself as a superior risk scoring system to both ASCVD or FRS (41, 42).

Reviewer 2 Report

This article, entitled “Discrimination between Obstructive Coronary Artery Disease and Cardiac Syndrome X in Women with Typical Angina and Positive Exercise Test; Utility of Cardiovascular Risk Calculators”, assessed the risk scores of ASCVD, Framingham risk score (FRS) and SCORE between women with obstructive coronary artery disease and cardiac syndrome. Authors reported that a low score of ASCVD or FRS was associated with a low risk of obstructive coronary artery disease in women with typical angina. Authors concluded that patients who have ASCVD < 2.5% or FRS < 7.5% may be appropriate cases for noninvasive imaging. This study is well designed and the article is well written. However, some questions should be clarified.

Major queries

SCORE system was developed based on European cohort data sets, and the risk calculator is only suitable for healthy persons without clinical disease. For example, diabetes will increase the CAD risk which cannot be shown by SCORE system. Since a significantly higher proportion of patients with diabetes in CAD group than CSX group, authors should do some analyses to exclude the confounding.

There is an over speculation in the conclusion. Authors did not evaluate the power of noninvasive imaging in differentiation between CAD and CSX. It is not suitable to suggest noninvasive imaging for women with typical angina, even with ASCVD < 2.5% or FRS < 7.5%.  

Minor queries

Authors should clearly describe how to choose the cutoff points for the risk scores. Were they selected based on ROC analysis?

The proposed algorithm to differentiate CAD from CSX should be shown in a figure for easy understanding.

Author Response

major comment 1: corrected in results:

Results of Table 3 indicate to markedly increased multivariate association of scoring models with obstructive CAD with respect to increased predicted risk. For instance, individuals who had moderate and high-risk profile according to ASCVD attained Odds ratios about 14.5 and 3.7 (compared with low risk category). We also found significant interdependences pertaining to the other models. A subgroup analysis for women according to diabetes showed that high SCORE points (risk ≥ 1.5 % vs < 0.5 %) are associated with greater likelihood of CAD. Corresponding odds ratios were 11.42 (2.54-16.71) and 5.53 (1.43- 8.92) for diabetics and non-diabetics respectively. Comparison of intermediate and low risk patients ((0.5-1.5 %) against < 0.5 %) did not lead to statistical significant results.

major comment 2: overspeculation was correctected (adjusted) 

abstract:

CAG is indicated for patients with ASCVD ≥ 7.5% and FRS ≥ 15% whereas the patients with intermediate risk need comprehensive patients-physician shared decision-making.

conclusions:

Patients who have ASCVD < 2.5 percentage and those who have FRS < 7.5 percentage may be appropriate cases for CTA instead of CAG. However, we must consider the crucial principles of clinical approach to patients with chest discomfort. These scoring models are secondary tools, which may help decision-making. Thus, in women who have clinical features like warning ECG changes, typical recurrent refractory unstable angina, are suitable for CAG in spite of having low scores. 

minor comment 1:

cut points were  determined according to values found in the reference studies of these scoring models and/ or quartiles of  our study population.

minor comment 2:

 Suggestion of an algorithm was not the main goal of the present investigation.However, it will be a good idea to design a prospective study to test the utility of different scores in step-wise diagnosis and management of women with angina.

Round 2

Reviewer 2 Report

All of my questions have be well responsed.

Author Response

All recommended changes were applied and ethical code was placed in the Methods section.
